# Decoupling Knowledge and Context: An Efficient and Effective Retrieval Augmented Generation Framework via Cross Attention

## Abstract

Retrieval-Augmented Generation (RAG) systems have become a crucial tool to augment large language models (LLMs) with external knowledge for better task performance. However, existing traditional RAG methods inject knowledge directly in the context, resulting in several limitations. First, these methods highly rely on the in-context learning capability of LLMs, which often leads to excessively long contexts. This is inefficient due to the quadratic complexity of self-attention, leading to significant increases in inference time. Second, the extended context and the nature of self-attention can cause the LLMs to lose important information in the context, thereby degrading the original capabilities of LLMs. Furthermore, the effectiveness of knowledge injection is perturbed by the permutation of knowledge within the extended context, reducing the robustness of existing RAG methods. To tackle the above problems, we propose **DecoupledRAG**, a method that decouples external knowledge from the context within the RAG framework. Specifically, we introduce a cross-attention based method that injects retrieved knowledge directly to the inference process of LLM on the fly, without modifying its parameters or the input context. The external knowledge could be utilized robustly in a permutation-independent manner. To the best of our knowledge, this is the first work that explore how to utilize cross-attention to inject knowledge with low training cost in decoder-only LLM era. By leveraging cross-attention operation, DecoupledRAG enables seamless knowledge aggregation without creating extended context. Experimental results demonstrate that our method achieves high efficiency while maintaining strong performance, which indicates that RAG frameworks have the potential to benefit further from more knowledge [1] [2].

## CCS Concepts

• **Information systems → Retrieval tasks and goals**.

## Keywords

Retrieval Augmented Generation, Language Model, Knowledge Injection

---

[1]The codes are released at https://anonymous.4open.science/r/DecoupledRAG

[2]Our work is related to "Search and retrieval-augmented AI" track as it contributes to optimize retrieval-augmented generation system.

---

**ACM Reference Format:**

Anonymous Author(s). 2018. Decoupling Knowledge and Context: An Efficient and Effective Retrieval Augmented Generation Framework via Cross Attention. In *Proceedings of Make sure to enter the correct conference title from your rights confirmation emai (Conference acronym 'XX).* ACM, New York, NY, USA, 10 pages. https://doi.org/XXXXXXX.XXXXXXX

## 1 Introduction

Retrieval-Augmented Generation (RAG) has emerged as a powerful approach for enhancing the capabilities of large language models (LLMs) by injecting external knowledge into their context. RAG enables LLMs to generate correct and timely responses based on knowledge retrieved from external corpus that may not be presented to the models in their training process, significantly boosting their performance across a range of knowledge-intensive natural language processing (NLP) tasks [17].

Despite the benefits, existing RAG methods face several issues. As shown in Figure 2(a), VanillaRAG (i.e., in-context manner) directly concatenates the retrieved external knowledge with the instruction and question. Although VanillaRAG can help mitigate the issue of incorrect or outdated responses, this approach inevitably leads to excessively long context, reducing both effectiveness and efficiency. The reasons lies in the following aspects. First, due to the quadratic complexity of self-attention [26], the processing of extended context substantially increases inference time. Second, the extended context and the nature of self-attention can cause LLMs to lose important information in the context [11, 14, 20], thereby degrading the original capabilities of LLMs, as demonstrated in Appendix A. Furthermore, due to the *lost in the middle* issue [11, 20], the effectiveness of knowledge injection could be perturbed by the permutation of knowledge within the extended context, reducing the robustness of VanillaRAG. Therefore, as shown in our preliminary experiment (see Figure 1), increasing the number of knowledge documents injected into the context significantly reduces the performance of LLMs on the T-REx [7] dataset. The issues of long texts limit the VanillaRAG framework from utilizing extensive useful knowledge efficiently and effectively [11, 14, 20]. Therefore, a natural research question is: **Could we construct a new RAG paradigm in which the injected knowledge and context are decoupled?**

Decoupling knowledge from context offers several advantages in the RAG system. First, it enhances efficiency by enabling the offline caching of external knowledge, allowing the LLM to efficiently utilize knowledge representations during inference on the fly. Second, decoupling mitigates the information loss issues that often arise in long contexts [11, 14, 20]. In this work, we propose utilizing cross-attention to inject the knowledge representations into the LLM, thereby decoupling the external knowledge from the context. The knowledge could be utilized in a permutation-independent manner via cross-attention, further alleviating the information loss issue [20]. To the best of our knowledge, this is the first work that

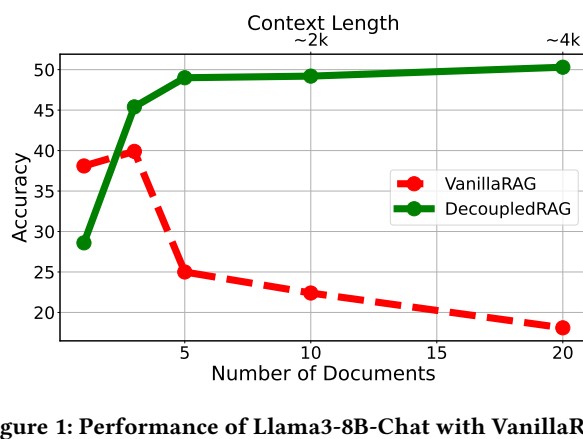

**Figure 1: Performance of Llama3-8B-Chat with VanillaRAG and DecoupledRAG on T-REx. The maximum context length of this LLM is 8k.**

explore how to utilize cross-attention to inject knowledge with low training cost in decoder-only LLM era.

However, equipping LLMs with the capability to inject external knowledge via cross-attention is non-trivial due to the following two challenges:

- **Challenge 1.** The training cost should not be excessively high. LLMs already contain extensive parameters, resulting in inherently high computational overhead. Retraining the entire LLM or introducing excessive additional trainable parameters is impractical.
- **Challenge 2.** Equipping the LLM with this capability should not collapse its original capabilities. This requires that training should build on the LLM's existing capabilities incrementally, ensuring they remain unaffected.

To tackle these challenges, our DecoupledRAG framework divides the workflow into two stages: knowledge encoding and knowledge aggregation, as shown in Figure 2(b). For *knowledge encoding*, we use the same LLM to pre-compute and cache representations of external knowledge, ensuring alignment with the context representations. This alignment allows the LLM to effectively utilize these pre-computed knowledge representations with minimal additional training overhead, thereby addressing **Challenge 1**. For *knowledge aggregation*, the cached representations of the external knowledge are retrieved from the database and integrated with the next token's representation via cross-attention, producing the next token representation with respect to external knowledge. Simultaneously, the next token undergoes a self-attention operation with the context to generate the representation with respect to internal context. We then apply coordinate-wise summation of these representations, with the weight for external knowledge initially set to zero. This setting ensures that injecting external knowledge does not disrupt the LLM's existing capabilities at the begin of training, addressing **Challenge 2**. As training progresses, LLM learns to balance internal and external knowledge appropriately. To further reduce the introduced parameters and maintain a low training cost, we decompose the zero matrix used in representation summation into two low-rank matrices. This parameter-efficient approach further addresses **Challenge 1**. We conduct extensive experiments across

multiple tasks to validate the effectiveness of our method. The experimental results demonstrate that our approach achieves high efficiency while maintaining strong performance, paving the way for more efficient and effective RAG systems.

To sum up, our contributions lies in the flowing aspects:

- We introduce DecoupledRAG, a novel RAG paradigm in which the injected knowledge and context are decoupled.
- We propose a low-cost training method that enables LLMs to effectively inject external knowledge via cross-attention without compromising their original capabilities.
- We conduct comprehensive experiments across multiple tasks, demonstrating the high efficiency and strong performance of our method.

## 2 Related Works

### 2.1 Large Language Models

Large Language Models (LLMs) have become a foundational component in natural language processing (NLP), achieving remarkable results in various tasks. One of the earliest milestones is the introduction of Transformer [26], which revolutionize NLP by introducing attention mechanism. This architecture pave the way for models such as BERT [5], GPT [21], and T5 [22], which have demonstrated state-of-the-art performance across tasks like question answering, summarization, and translation.

As model sizes grow, LLMs like GPT-3 [3] and PaLM [4] show the power of scaling, with billions of parameters enabling models to generalize better to a variety of prompts. Despite their strengths, LLMs rely on static knowledge acquired during pre-training, which limits their ability to adapt to real-time information. Additionally, hallucination remains a concern, where the model generates confident but incorrect or nonsensical outputs, highlighting the need for external knowledge augmentation.

### 2.2 Knowledge Injection in Large Language Models

Several recent methods have been proposed to enable effective knowledge injection into LLMs. One common strategy involves the use of Supervised Fine-Tuning (SFT), where out-of-domain knowledge is injected by fine-tuning models like Llama-3 [1] with new datasets. This approach has demonstrated significant improvements in question answering accuracy, particularly in domains where the model's pre-existing knowledge is insufficient. Other approaches, such as those explored by KnowGPT [29], inject external knowledge from structured sources like knowledge graphs into LLMs, helping mitigate hallucination and improve factual consistency. Moreover, Graph-Reader [19] utilizes graph-based knowledge representations to enhance long-context reasoning and knowledge injection in LLMs, further pushing the limits of context comprehension and factuality. These methods enable more precise and structured knowledge integration, providing models with the ability to reference external data efficiently. Retrieval-augmented generation (RAG) [18] has garnered significant attention in recent years. It extends knowledge injection by dynamically retrieving relevant documents during the generation process, making it a powerful tool for enhancing the factual accuracy and relevance of LLM outputs.

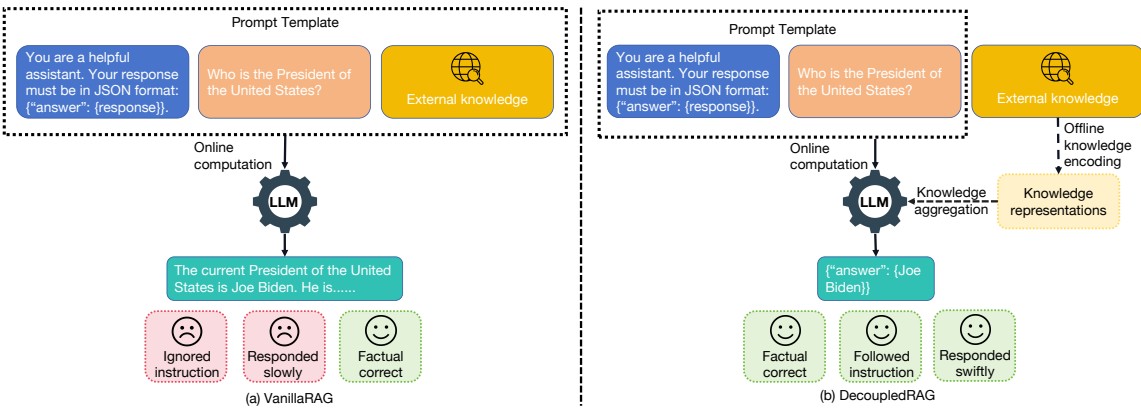

Figure 2: Comparison between VanillaRAG and our proposed DecoupledRAG.

## 2.3 Retrieval-Augmented Generation

RAG is initially introduced by this study [18], combining the capabilities of information retrieval and generative language models. The original RAG architecture uses a Dense Passage Retrieval (DPR) [12] model to index documents and a sequence-to-sequence model (BART) to generate responses [16].

DPR [12] is a foundational retrieval method that enables precise retrieval from large knowledge corpus like Wikipedia, and serves as the retrieval backbone for extensive RAG models [8, 18]. RAG leverages DPR to retrieve relevant documents, conditioning generation on these documents, and improving generation quality.

Recent agent-based RAG systems, such as PaperQA [13], Graph-Reader [19], and PersonaRAG [28], introduce specialized frameworks that combine retrieval and generation with agent-based capabilities. These systems aim to improve performance in long-context handling, reduce hallucination, and adapt to real-time user data.

## 3 Methodology

### 3.1 Preliminaries

Before delving into the details of our proposed method, we provide a formal definition of self-attention operation, as it plays a crucial role in both the knowledge encoding and aggregation stages. Self-attention is used in LLM to compute hidden state for generating the next token. Given a context $X = [x_1, x_2, \ldots, x_n]$, the self-attention operation computes attention scores to determine the importance of each token relative to its preceding tokens, producing a contextualized hidden state. For an LLM with $L$ layers, at each layer $l \in \{1, 2, \ldots, L\}$, the output of the previous layer $X^{(l-1)}$ serves as the input to the current layer, where $X^{(0)}$ represents the output of the embedding layer. To ensure that the model captures the sequential information of the context, positional encoding operation $(\text{Pos}(\cdot))$ is incorporated into the attention mechanism. The attention mechanism itself is invariant to the ordering of tokens, which means that without $\text{Pos}(\cdot)$, the model would have no information about the position of each token. Commonly used $\text{Pos}(\cdot)$ functions include sinusoidal positional encoding [26], learnable positional encoding [5], and rotary position embedding (RoPE) [23]. Formally, the self-attention output at layer $l$ is computed as

$$X^{(l)} = \text{softmax}\left(\frac{\text{Pos}(Q^{(l)})\text{Pos}(K^{(l)T})}{\sqrt{d_k}}\right)V^{(l)}, \quad (1)$$

where the query, key, and value matrices $Q^{(l)}, K^{(l)}$, and $V^{(l)}$ at layer $l$ are computed from the hidden states of the previous layer $X^{(l-1)}$. Specifically, these matrices could be computed as

$$Q^{(l)} = X^{(l-1)}W_Q^{(l)}, \quad K^{(l)} = X^{(l-1)}W_K^{(l)}, \quad V^{(l)} = X^{(l-1)}W_V^{(l)}. \quad (2)$$

Here, $W_Q^{(l)}, W_K^{(l)}$, and $W_V^{(l)}$ are learned projection matrices at layer $l$ that map the hidden states of the input $X^{(l-1)}$ into the query, key, and value spaces, respectively. After processing through all $L$ layers of the LLM, the hidden state from the final layer of last token $x_n^{(L)}$ is used to generate the next token. Formally, the next token $x_{n+1}$ is obtained by

$$x_{n+1} = \text{argmax}(\text{softmax}(x_n^{(L)}W_O + b_O)), \quad (3)$$

where $W_O$ is the output projection matrix, and $b_O$ is the bias term. The softmax function is applied to produce a probability distribution over the vocabulary, allowing the model to predict the next token based on the final layer's hidden state. After generating $x_{n+1}$, it is added back into the context, i.e., $X = [x_1, x_2, \ldots, x_{n+1}]$.

VanillaRAG methods directly inject knowledge documents into the input context to improve the generation quality. Given a question $Q$ and a set of corresponding retrieved documents $\mathcal{D}^Q = \{D_1^Q, D_2^Q, \ldots, D_N^Q\}$, the input is formed by concatenating the question and the documents through a template $\mathcal{T}$, denoted as

$$X = \mathcal{T}(\mathbb{T}, Q, \mathcal{D}^Q), \quad (4)$$

where $\mathbb{T}$ represents task-specific instructions. The structure of the template $\mathcal{T}$ and the task-specific instructions $\mathbb{T}$ are defined in the Appendix B. Then, the LLM auto-regressively predicts each next token one by one using Eq. 3. This process is repeated for each subsequent token until the complete answer is generated.

Notably, the main challenges in equipping LLM with the ability to utilize cross-attention for knowledge injection are: (1) training the LLM efficiently and (2) preserving its original capabilities. We tackle these challenges by two stages in DecoupledRAG. The framework of DecoupledRAG is depicted in Figure 3, which includes tow stages,

*knowledge encoding* and *knowledge aggregation*. Next, we present the details of each stage of DecoupledRAG.

## 3.2 Knowledge Encoding

The knowledge encoding stage pre-computes the representations of external knowledge for use in subsequent knowledge aggregation in an on-the-fly manner. To reduce training difficulty, we ensure the compatibility between the external knowledge representations and the internal context representations. Therefore, we use the same LLM to encode external knowledge.

To formally define the knowledge encoding stage, we denote $D = [d_1, d_2, \ldots, d_m]$ as a external knowledge sequence, where $D$ is consist of $m$ tokens. At layer $l$ of an LLM, the hidden states $D^{(l)}$ is computed from the hidden states of the previous layer $D^{(l-1)}$ by Eq. 1. Then, we store the key-value representations of the external knowledge. Specifically, at each layer $l$, the key and value representations $K_D^{(l)}$ and $V_D^{(l)}$ are computed as

$$K_D^{(l)} = D^{(l-1)} W_K^{(l)}, \quad V_D^{(l)} = D^{(l-1)} W_V^{(l)}, \tag{5}$$

where $W_K^{(l)}$ and $W_V^{(l)}$ are the learned projection matrices from the same LLM. The cached key-value representations for all layers are stored as

$$\mathcal{K}_D = \{(K_D^{(l)}, V_D^{(l)})\}_{l=1}^L. \tag{6}$$

Then, the cached key-value representations, $\mathcal{K}_D$, can be directly used for knowledge aggregation. It is worth noting that when a LLM uses Grouped-Query Attention [2] for acceleration, we only need to store the key-value representations for each group, significantly reducing memory overhead.

## 3.3 Knowledge Aggregation

Once the external knowledge is encoded and cached, the next step is to inject it into the LLM through cross-attention. Since we may inject multiple external knowledge, we concatenate all the key-value representations of external knowledge before performing the cross-attention.

Specifically, the context used in DecoupledRAG can be formed as

$$X = \mathcal{T}(\mathbb{T}, Q), \tag{7}$$

which decouples the external knowledge $\mathcal{D}^Q$ from the instruction $\mathbb{T}$ and the question Q. Since the subsequent formulas all focus on a specific question Q, we omit the subscript Q in $\mathcal{D}^Q$ for clarity. For multiple external knowledge $\mathcal{D} = \{D_1, D_2, \ldots, D_N\}$, we concatenate the key-value representations at layer $l$ of all $N$ external knowledge sequences as

$$K_{ext}^{(l)} = [K_{D_1}^{(l)}, K_{D_2}^{(l)}, \ldots, K_{D_N}^{(l)}], \quad V_{ext}^{(l)} = [V_{D_1}^{(l)}, V_{D_2}^{(l)}, \ldots, V_{D_N}^{(l)}]. \tag{8}$$

For the aggregation process, the last token's hidden state is first integrated through the *self-attention* operation with the internal context representations, followed by integration with the concatenated external knowledge representations through the *cross-attention* operation. Formally, the *self-attention* operation for the last token $x_n$ at layer $l$ can be defined as

$$x_{n,\text{int}}^{(l)} = \text{softmax}\left(\frac{\text{Pos}(Q_{x_n}^{(l)})\text{Pos}(K^{(l)T})}{\sqrt{d_k}}\right) V^{(l)}, \tag{9}$$

$x_{n,\text{int}}^{(l)}$ is the token representation with respect to internal context. Next, the representation of the last token $x_n$ undergoes a *cross-attention* operation with the concatenated external knowledge. This step aggregates information from the external knowledge to produce the token representation $x_{n,\text{ext}}^{(l)}$, which could be defined as

$$x_{n,\text{ext}}^{(l)} = \text{softmax}\left(\frac{Q_{x_n}^{(l)} K_{ext}^{(l)T}}{\sqrt{d_k}}\right) V_{ext}^{(l)}. \tag{10}$$

Notably, the knowledge aggregation is *permutation-independent* due to the absence of $\text{Pos}(\cdot)$ function, enabling DecoupledRAG to inject knowledge without considering the order of knowledge documents.

Finally, the hidden state of the last token at layer $l$ is obtained by combining the self-attention and cross-attention outputs, which can be computed as

$$x_n^{(l+1)} = x_{n,\text{int}}^{(l)} + W_\beta^{(l)} x_{n,\text{ext}}^{(l)}, \tag{11}$$

where $W_\beta^{(l)}$ is introduced as a learnable weight matrix to control the influence of external knowledge during the aggregation of internal and external hidden states. We initialize $W_\beta^{(l)}$ as a zero matrix, ensuring that at the start of training, LLM relies entirely on its internal knowledge, with the contribution from external knowledge gradually learned during fine-tuning. This initialization prevents the collapse of the LLM's original capabilities and facilitates the smooth aggregation of external knowledge. Zero initialization of $W_\beta^{(l)}$ is crucial because if external knowledge is weighted too heavily at the start of training, it could significantly disrupt the LLM's behavior. This would necessitate a complete retraining of the model, introducing excessive training costs.

Inspired by low rank adaptation [10], we further reduce the number of trainable parameters in $W_\beta$ by decomposing it into two low-rank matrices and a scaling factor

$$W_\beta^{(l)} = \alpha A_\beta^{(l)} B_\beta^{(l)}, \tag{12}$$

where $A_\beta^{(l)} \in \mathbb{R}^{d \times r}$ and $B_\beta^{(l)} \in \mathbb{R}^{r \times d}$ are low-rank matrices, with $r \ll d$. $A_\beta^{(l)}$ is initialized with Gaussian noise, while $B_\beta^{(l)}$ is initialized as a zero matrix. This decomposition enables the model to learn how to integrate external knowledge efficiently, with minimal additional training costs. Therefore, the Eq. 11 could be rewritten as

$$x_n^{(l+1)} = x_{n,\text{int}}^{(l)} + \alpha A_\beta^{(l)} B_\beta^{(l)} x_{n,\text{ext}}^{(l)}. \tag{13}$$

The next token is obtained in the same manner as in Eq. 3.

## 3.4 Model Training

Both VanillaRAG and our DecoupledRAG are optimized using the next token prediction objective, commonly employed for auto-regressive language models training.

Following recent work [11], VanillaRAG is trained using the standard next token prediction objective with teacher forcing, where the model is provided with the ground truth answer during training. The input to the model is represented as $S = X + A$, where $X = \mathcal{T}(\mathbb{T}, Q, \mathcal{D})$ and $A$ is the ground truth answer. Formally, the

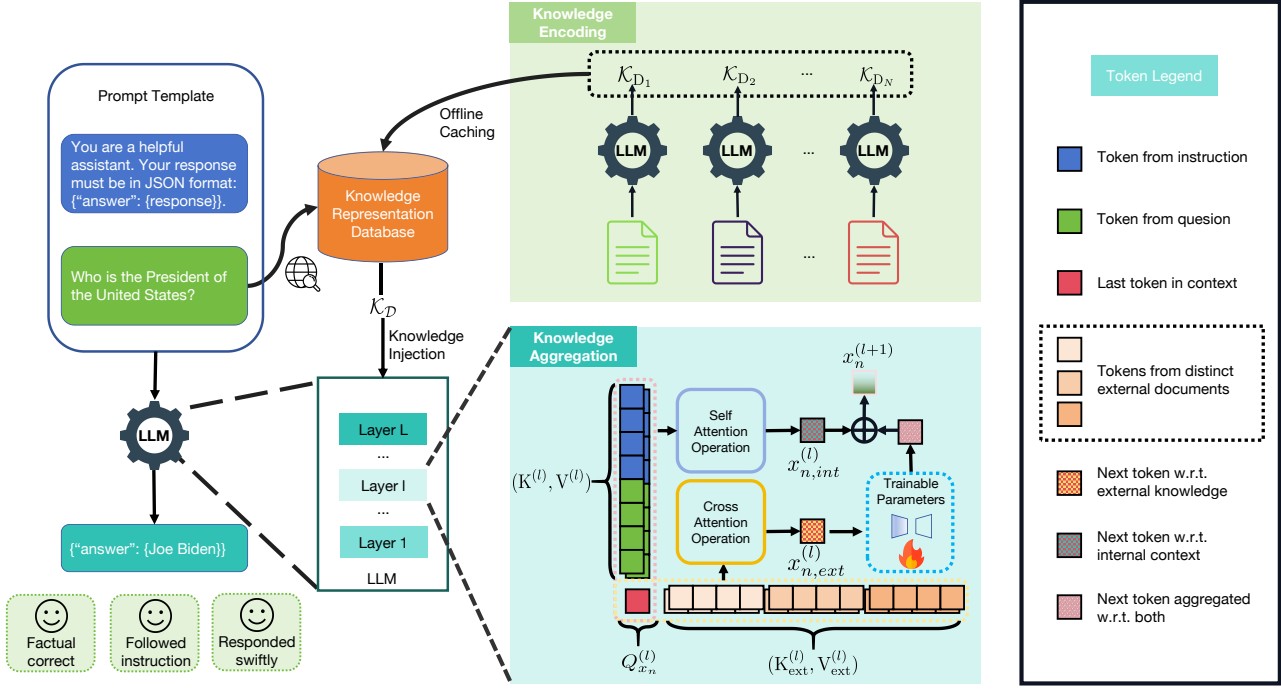

**Figure 3: The illustration of DecoupledRAG framework.**

training objective is to minimize the cross-entropy loss as follows

$$\mathcal{L} = -\sum_{i=1}^{|S|} \log P_\theta(y_{i+1} = s_{i+1} | s_1, s_2, \ldots, s_i), \quad (14)$$

where $y_{i+1}$ represents the predicted token.

In contrast, our DecoupledRAG follows the same next token prediction objective but decouples the external knowledge from the input context, i.e., $S = X + A$, where $X = \mathcal{T}(\mathbb{T}, Q)$. While the training objective remains a next-token prediction task, DecoupledRAG generates the next token based on preceding tokens and pre-cached knowledge representations $\mathcal{K}_\mathcal{D} = \{\mathcal{K}_{D_1}, \mathcal{K}_{D_2}, \ldots, \mathcal{K}_{D_N}\}$, which can be denoted as

$$\mathcal{L} = -\sum_{i=1}^{|S|} \log P_\theta(y_{i+1} = s_{i+1} | \mathcal{K}_\mathcal{D}, s_1, s_2, \ldots, s_i). \quad (15)$$

## 3.5 Computational Complexity

In this subsection, we analyze the computational complexity of aggregating external knowledge using self-attention and cross-attention operations, respectively. Let $|D|$, $|Q|$, and $|A|$ denote the number of tokens in the external knowledge, question, and answer, respectively.

**Self-Attention for External Knowledge Aggregation.** When using self-attention to aggregate external knowledge, the external knowledge is treated as part of the context, together the question and answer. This results in the following computational complexity

$$O\left( (N \cdot |D| + |Q|)^2 + |A| \cdot (N \cdot |D| + |Q|) \right), \quad (16)$$

where $N$ represents the number of external knowledge. Since $|Q| \ll |D|$, the Equation (16) could be simplified as

$$O\left( \overbrace{(N \cdot |D|)^2}^{\text{Knowledge encoding}} + \overbrace{|A| \cdot (N \cdot |D|)}^{\text{Answer generation}} \right). \quad (17)$$

$$\underbrace{\phantom{(N \cdot |D|)^2 + |A| \cdot (N \cdot |D|)}}_{\text{Online inference cost}}$$

As $N$ increases, the online inference cost of VanillaRAG grows exponentially.

**Cross-Attention for External Knowledge Aggregation.** In contrast, DecoupledRAG encodes knowledge documents independently from each other and from the question. This reduces the computational complexity, which can be expressed as

$$O\left( N \cdot |D|^2 + |Q|^2 + |A| \cdot (N \cdot |D| + |Q|) \right). \quad (18)$$

Similarly, this equation could be simplified as

$$O\left( \overbrace{N \cdot |D|^2}^{\text{Knowledge encoding}} + \overbrace{|A| \cdot (N \cdot |D|)}^{\text{Answer generation}} \right). \quad (19)$$

$$\underbrace{\phantom{N \cdot |D|^2}}_{\text{Offline inference cost}} \quad \underbrace{\phantom{|A| \cdot (N \cdot |D|)}}_{\text{Online inference cost}}$$

Here, the quadratic term $N \cdot |D|^2$ reflects the offline cost of encoding each external knowledge separately. This results in more efficient handling of larger external knowledge sets, as the question and external knowledge are all decoupled. Notably, the encoded knowledge is question-independent and can therefore be used for all questions, rather than being tied to a specific one.

As $N$ increases, DecoupledRAG demonstrates increasingly superior efficiency compared to VanillaRAG, since the online inference cost of DecoupledRAG grows linearly.

## 4 Experiments

In this section, we present the experiments conducted to evaluate the performance of our proposed DecoupledRAG. We first describe the datasets and evaluation metrics used, followed by details of the experimental setup. Finally, we present the results and provide a comprehensive analysis.

### 4.1 Datasets and Metrics

We evaluate our proposed method across three distinct tasks, using five datasets to comprehensively assess performance.

**Multi-hop Question Answering.** 2WikiMultihopQA [9] is designed to test the multi-hop reasoning capabilities across multiple Wikipedia articles, requiring models to gather and synthesize information from multiple sources to answer complex questions. The ComplexWebQuestions [24] dataset involves answering multi-step, web-based questions, further challenging the LLM's ability to retrieve and reason over large-scale web content. We use accuracy and F1 score to evaluate these tasks. Accuracy is measured as the percentage of answers that exactly match the ground truth (EM) divided by the total number of questions in the test set, while the F1 score captures the balance between precision and recall by accounting for partially correct answers.

**Slot Filling.** For slot-filling tasks, we evaluate our method on the Zero-Shot RE [15] and T-REx [7] datasets. Zero-Shot RE [15] is used for zero-shot relation prediction, where LLMs are tested on relations they haven't explicitly seen in training. T-REx is a large-scale factual knowledge dataset used to evaluate the LLM's ability to fill in factual slots using external knowledge from Wikipedia. For these datasets, we use both accuracy and F1 score.

**Dialogue.** We use the Wizard of Wikipedia (WoW) [6] dataset for dialogue tasks. In this dataset, the LLM is expected to engage in knowledgeable conversations by leveraging external knowledge retrieved from Wikipedia articles. The task tests both the knowledge integration capabilities of the LLM and its ability to maintain coherent dialogue. For this task, we use F1 score to evaluate the LLM's ability to generate relevant and correct responses during conversations.

### 4.2 Experimental Setup

We implement our approach on top of pre-trained LLMs, specifically *Llama3-8B-Instruct* [1] and *Llama2-7B-Chat* [25], using the Hugging Face Transformers library. Both training and evaluation take place on up to 8 NVIDIA A100 GPUs with 40GB of memory. The training process runs for 5 epochs, with a learning rate of 1e-3 and a batch size of 16.

We use Wikipedia as the knowledge corpus. Documents are divided into non-overlapping segments, each consisting of exactly 256 tokens. Any final segment with fewer than 128 tokens is discarded. After segmentation, the corpus comprises approximately 21 million knowledge documents. For the experiments, 1, 3, 5, 10, and 20 knowledge documents are injected to assess performance,

respectively. We use RetroMAE [27] as the retrieval module to recall relevant external knowledge. In each experiment, all baselines and our method utilize the same set of retrieved documents to ensure a fair comparison and consistent evaluation across different approaches.

During LLM fine-tuning, we apply LoRA with rank $r = 16$ and $\alpha = 32$, introducing a total of 6.82M additional parameters. Similarly, our proposed DecoupledRAG also employs $r = 16$ and $\alpha = 32$ for Eq. 13, resulting in an 4.19M additional parameters.

### 4.3 Experimental Results

Table 1 compares the performance of our proposed DecoupledRAG method with VanillaRAG across three tasks: Slot Filling, Multi-hop Question Answering, and Dialogue. We evaluate the effectiveness of both methods by injecting 1, 3, 5, 10, and 20 external knowledge documents. From this table, we can draw the following findings:

- DecoupledRAG demonstrates superior performance with more knowledge documents injected. This performance gain can be attributed to DecoupledRAG's ability to avoid increasing the context length while effectively aggregating external knowledge. Consequently, the LLM benefits from external knowledge to generate accurate responses while preserving its instruction following capability.
- When a limited number of external knowledge documents are injected, VanillaRAG performs slightly better than DecoupledRAG. The reason lies in that self-attention provides more comprehensive interaction across the entire context. The knowledge representations could aggregate information from the instruction and the question, making LLM can effectively follow the instruction and focus on the question.
- VanillaRAG requires a trade-off between the number of external knowledge and the length of internal context. The optimal number of injected knowledge documents varies across different datasets and models. For example, with Llama-3-8B-Instruct, the best performance is observed with two injected documents in Zero-Shot RE, T-REx, and 2WikiMultihopQA, while three documents yield the highest results in ComplexWebQuestions. In WoW, VanillaRAG achieves best performance with just one injected document.
- Overall, Llama-3-8B-Instruct outperforms Llama2-7B-Chat consistently across all datasets, owing to its stronger foundational capabilities and enhanced ability to handle longer contexts effectively.

To facilitate the analysis of trends, we present Figure 4 and 5, which compare the performance of DecoupledRAG and VanillaRAG across multiple datasets using Llama-3-8B-Instruct and Llama-2-7B-Chat, respectively. Since the trends for Accuracy and F1 are consistent in the Slot Filling and Multi-hop Question Answering tasks, we only present the Accuracy results. From these figures, we can draw several key observations:

- Compared to VanillaRAG, DecoupledRAG shows a steady improvement in performance as more external knowledge documents are injected. This upward trend stems from DecoupledRAG's ability to decouple external knowledge from the context, ensuring that injecting additional knowledge does not overwhelm the important information in the

**Table 1: Performance comparison between VanillaRAG and DecoupledRAG. The best performances are highlighted in bold.**

| # Docs | Method | Slot Filling | | | | Multi-hop Question Answering | | | | Dialogue |
|---|---|---|---|---|---|---|---|---|---|---|
| | | Zero-Shot RE | | T-REx | | 2WikiMultihopQA | | ComplexWebQuestions | | WoW |
| | | Acc. | F1 | Acc. | F1 | Acc. | F1 | Acc. | F1 | F1 |
| | | Llama-3-8B-Instruct | | | | | | | | |
| +1 doc | VanillaRAG | 38.1 | 49.9 | 61.2 | 63.2 | 17.2 | 12.1 | 23.4 | 31.9 | 22.3 |
| | DecoupledRAG | 28.6 | 37.1 | 61.1 | 63.9 | 15.2 | 11.1 | 9.8 | 16.8 | 24.4 |
| +3 doc | VanillaRAG | 39.9 | 51.3 | 69.3 | 72.8 | 29.5 | 34.3 | 15.3 | 21.3 | 22.0 |
| | DecoupledRAG | 45.4 | 53.1 | 73.7 | 76.2 | 28.1 | 32.5 | 29.2 | 37.0 | 25.5 |
| +5 doc | VanillaRAG | 25.0 | 29.9 | 35.7 | 37.4 | 26.3 | 31.4 | 10.1 | 17.5 | 18.9 |
| | DecoupledRAG | 49.0 | 56.3 | 76.3 | 78.3 | 29.9 | 34.3 | 33.8 | 41.8 | 25.9 |
| +10 doc | VanillaRAG | 22.4 | 28.0 | 30.2 | 32.0 | 13.1 | 23.4 | 12.5 | 19.6 | 19.3 |
| | DecoupledRAG | 49.2 | 56.9 | 78.4 | 80.3 | 32.7 | 37.6 | 37.6 | 45.1 | 25.3 |
| +20 doc | VanillaRAG | 18.1 | 24.8 | 25.3 | 27.4 | 3.8 | 9.6 | 15.7 | 24.0 | 19.3 |
| | DecoupledRAG | **50.3** | **57.5** | **80.2** | **81.9** | **33.4** | **38.1** | **39.6** | **47.1** | **26.1** |
| | | Llama-2-7B-Chat | | | | | | | | |
| +1 doc | VanillaRAG | 3.7 | 5.1 | 18.1 | 19.2 | 13.1 | 18.9 | 11.0 | 14.7 | 18.5 |
| | DecoupledRAG | 2.2 | 4.1 | 11.8 | 13.0 | 13.2 | 19.3 | 10.5 | 14.0 | 17.3 |
| +3 doc | VanillaRAG | 3.1 | 4.6 | 18.3 | 19.5 | 15.5 | 20.7 | 12.5 | 15.9 | 19.2 |
| | DecoupledRAG | 4.2 | 5.6 | 17.4 | 18.7 | 16.6 | 21.8 | 12.2 | 15.8 | 17.3 |
| +5 doc | VanillaRAG | 3.4 | 4.8 | 17.7 | 18.9 | 12.2 | 16.5 | 9.8 | 12.4 | 15.6 |
| | DecoupledRAG | 4.2 | 5.5 | 19.0 | 20.1 | 17.4 | 22.6 | 13.4 | 16.7 | 17.8 |
| +10 doc | VanillaRAG | 2.5 | 4.2 | 6.2 | 7.4 | 6.5 | 8.2 | 5.1 | 6.3 | 7.4 |
| | DecoupledRAG | 4.6 | 5.9 | 18.5 | 19.7 | 17.7 | 23.1 | 13.1 | 16.5 | 18.2 |
| +20 doc | VanillaRAG | 1.7 | 3.8 | 6.9 | 7.3 | 3.2 | 4.0 | 2.3 | 3.0 | 3.6 |
| | DecoupledRAG | **4.8** | **6.1** | **20.6** | **21.8** | **18.2** | **23.5** | **13.8** | **17.0** | **20.0** |

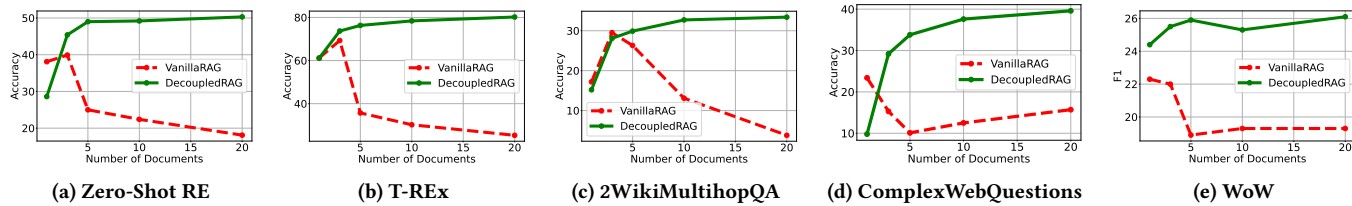

(a) Zero-Shot RE     (b) T-REx     (c) 2WikiMultihopQA     (d) ComplexWebQuestions     (e) WoW

**Figure 4: Performance of Llama-3-8B-Instruct across datasets**

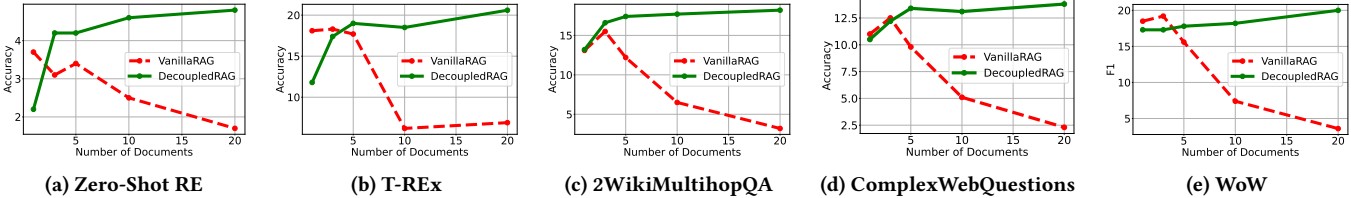

(a) Zero-Shot RE     (b) T-REx     (c) 2WikiMultihopQA     (d) ComplexWebQuestions     (e) WoW

**Figure 5: Performance of Llama-2-7B-Chat across datasets**

context. However, with few external documents, the performance of DecoupledRAG is limited due to cross-attention interactions being less comprehensive than self-attention.
- In contrast, VanillaRAG experiences a significant drop in performance as the number of injected documents increases, particularly in tasks like Zero-Shot RE and ComplexWebQuestions. This decline further demonstrates the self-attention mechanism in VanillaRAG is inefficient with longer contexts, hindering the LLM' capability to utilize important information in instruction and knowledge.
- DecoupledRAG consistently outperforms the highest performance of VanillaRAG across all datasets, demonstrating

its superior ability to effectively leverage knowledge without compromising the essential information in the context.

## 4.4 Comparison with Long-Context RAG Methods

The most relevant work to ours is a recent study [11] published by Google, which provides an in-depth analysis of the challenges faced by RAG w.r.t. long-context. The study also proposes three potential solutions to address these issues.

**Retrieval Reordering (RR).** This is a training-free method proposed to address the *lost in the middle* issue. By reordering the knowledge documents based on their relevance scores to the question, the most relevant documents are placed at the beginning and end of the context.

**RAG Fine-Tuning (RAG FT).** This method aims to improve LLM robustness to hard negatives by training them with retrieved knowledge documents, which is identical to the settings of VanillaRAG in our experiments.

**RAG FT with an intermediate reasoning (RAG FT w/. Int).** RAG FT w/. Int explicitly trains the LLM to differentiate between relevant and irrelevant passages within the retrieved context by generating a reasoning paragraph. However, the inference time overhead is significantly scaling up due to the additional reasoning generation, making the comparison unfair. Therefore, this method is not included in our comparison.

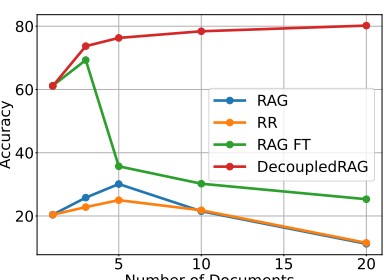

**Figure 6: Comparison of different long-context RAG methods on the T-REx dataset.**

The results are presented in Figure 6. Since LLMs are sensitive to the position of knowledge documents, RR exhibits unstable performance on the T-REx dataset, especially when the number of documents is limited, demonstrating that LLM performance is easily perturbed by the permutation of knowledge. RAG-oriented fine-tuning is an effective method for improving performance, but as the number of documents increases (e.g., greater than 5), performance declines sharply, which highlights the limitation of long context in LLMs. Notably, RAG FT is identical to VanillaRAG in our experiments. Our DecoupledRAG utilizes knowledge in a permutation-independent manner while avoiding excessively long contexts, thereby achieving superior performance.

## 4.5 Efficiency Analysis

In this subsection, we compare the online inference efficiency of DecoupledRAG and VanillaRAG in terms of Tokens per Second

(TPS). The results are presented in Figure 7. From this figure, we can draw the following conclusions:

- DecoupledRAG consistently demonstrates better efficiency in terms of TPS compared to VanillaRAG across two models, particularly as the number of injected documents increases. This is primarily because directly utilizing pre-computed knowledge representations in DecoupledRAG significantly reduces the computational overhead during inference.
- When a limited number of documents are injected (e.g., fewer than 5), the efficiency gap between DecoupledRAG and VanillaRAG is marginal. This is because the overhead introduced by knowledge encoding in VanillaRAG is less pronounced with a short context size. As demonstrated in Equation (17), when $N$ is less than 5, the knowledge could be encoded in a single forward-pass with negligible time overhead. However, as the number of documents increases, DecoupledRAG exhibits a significantly better efficiency, attributed to the decoupling of knowledge and context.
- As the number of injected documents increases (from 0 to 50), the TPS for VanillaRAG drops significantly, while DecoupledRAG maintains a much more gradual decline. Notably, the TPS of DecoupledRAG with 50 injected documents still outperforms that of VanillaRAG with just 20 injected documents, as illustrated by the green dashed line in Figure 7. This further paves the way for injecting more knowledge with RAG framework.

In conclusion, DecoupledRAG offers superior efficiency, particularly when handling a large number of knowledge documents. Its decoupling mechanism enables LLMs to maintain fast token generation speed, making them more scalable and better suited for scenarios requiring the injection of substantial external knowledge.

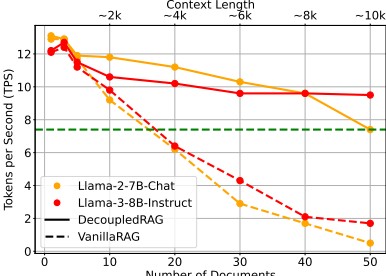

**Figure 7: Tokens per Second Performance Comparison.**

## 5 Conclusion

In this work, we present DecoupledRAG, a novel framework designed to address the inherent limitations of traditional context-based RAG systems. By decoupling external knowledge from the context and utilizing cross-attention for knowledge injection, DecoupledRAG mitigates the issues related to long context, such as increased inference latency and degradation of fundamental capabilities. Besides, DecoupledRAG is more robust to the permutation of knowledge, as the knowledge is injected in a permutation-independent manner. Extensive experiments across multiple datasets demonstrate that DecoupledRAG not only maintains high efficiency but also achieves superior performance.

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

SYSTEM: You are a helpful assistant. {$\mathbb{T}$}

USER: {$Q$}

Answer the question based on the references.
References: {$\mathcal{D}$}

ASSISTANT:

**Figure 8: The template $\mathcal{T}$ used in our experiments.**

**Table 2: The task-specific instructions.**

|  | Instruction |
|---|---|
| Zero-Shot RE | Please answer user question to the best of your ability. The answer MUST in ONE OR FEW WORDS. |
| T-REx | Please fill in the [MASK] in the sentence. |
| 2WikiMultihopQA | Answer the user question that require reasoning over multiple Wikipedia articles. The answer MUST in ONE OR FEW WORDS. |
| ComplexWebQuestions | Answer the user question that require reasoning over multiple Wikipedia articles. The answer MUST in ONE OR FEW WORDS. |
| WoW | Integrating knowledge from Wikipedia to improve the informativeness of your answers. |

## A  Failure Analysis for VanillaRAG

In this section, we present the failure analysis of VanillaRAG on the T-REx dataset using Llama-3-8B-Instruct. Specifically, we examine 400 randomly selected cases where the LLM provides correct answer with 1 injected document but fails when 20 documents are injected into its context. Through manual analysis, the failures can be categorized into three types:

- **Failure to Follow Instruction.** The T-REx instruction prompts the LLM to fill the [MASK] token in the given sentence based on the retrieved references. However, as the context length increases, the LLM often loses focus on the instruction and tends to either continue writing the context or summarize context. Among the 400 cases, 187 cases fall into this category.
- **Wrong Answer.** This issue arises because long context degrades the in-context learning capability of LLMs, even though the retrieved references are highly relevant to the question. LLM generates incorrect answers in 74 cases.
- **Meaningless Content.** Long contexts collapse the foundational capabilities of LLMs, resulting in the generation of random segments. This issue is observed in 139 cases.

From this analysis, it can be concluded that long contexts hinder the performance of VanillaRAG due to a decline in instruction-following, in-context learning, and other foundational capabilities in LLMs, further highlighting the limitations of directly injecting knowledge into the context.

## B  Template and Task-specific Instructions

The template $\mathcal{T}(\mathbb{T}, Q, \mathcal{D})$ used in our experiments is presented in Figure 8. The task-specific instructions are shown in Table 2.

Received 20 February 2007; revised 12 March 2009; accepted 5 June 2009

