# OpenReview forum: "Decoupling Knowledge and Context: An Efficient and Effective Retrieval Augmented Generation Framework via Cross Attention"
_ACM.org/TheWebConf/2025/Conference — WWW 2025 Poster_

### Official Review · Reviewer_AfAr · 2024-11-29

**Novelty:** 5
**Technical Quality:** 4

**Review:**

This paper addresses the issue of reduced inference efficiency and performance in existing retrieval-augmented generation (RAG) models caused by prompts containing an excessive number of documents. The authors propose a novel approach by separating the context and documents in the RAG model. This involves pre-computing key-value (KV) vectors of the documents, storing them, and directly integrating these vectors into the internal computation of the language model to merge document information. The results on LLaMA2-7B and LLaMA3-8B demonstrate the method's efficiency and effectiveness when handling multiple document inputs. However, the method has only been validated on limited models.

Pros:
1. The paper is generally well-organized and easy to follow.
2. The authors have made their code available in an anonymous repository, promoting transparency and enabling reproducibility of the research.

Cons:
1.  In Challenge 2, the paper mentions that "equipping the LLM with this capability should not collapse its original capabilities." However, the paper lacks experimental results to support that the proposed method achieves this.
2. The experiments were conducted using only the LLaMA series models. To effectively demonstrate the framework's innovation, it would be beneficial to include experiments with a broader variety of model series.
3. The experiments were only conducted on relatively small models (7B and 8B). It is unclear whether the issues addressed by the paper are equally severe in larger models, and whether the proposed solution remains effective at such scales.
4. It remains uncertain whether the phenomena addressed by the paper exist in more advanced models such as GPT-4 and those with extended context capabilities that feature input windows extending up to tens of thousands of characters.

**Questions:**

see the cons.

**Reviewer Confidence:**

3: The reviewer is confident but not certain that the evaluation is correct

**Scope:**

4: The work is relevant to the Web and to the track, and is of broad interest to the community

---

### Official Review · Reviewer_GdYS · 2024-11-30

**Novelty:** 5
**Technical Quality:** 5

**Review:**

### Contributions:

1. **DecoupledRAG Framework**: The paper introduces a novel Retrieval-Augmented Generation (RAG) framework called DecoupledRAG, which addresses the limitations of traditional RAG methods by decoupling external knowledge from context.

2. **Cross-Attention Mechanism**: It incorporates a cross-attention-based approach to inject retrieved knowledge directly into the inference process of Large Language Models (LLMs) without altering the model's parameters or input context.

3. **Permutation Independence**: DecoupledRAG robustly utilizes external knowledge in a permutation-independent manner, mitigating the issue of information loss and enhancing the robustness of the method.


### Weaknesses:

1. While DecoupledRAG aims to resolve issues stemming from long contexts, its performance may be limited in scenarios with few external documents due to less comprehensive interactions with cross-attention compared to self-attention.

2. The introduction of cross-attention and parameter decomposition may increase the complexity of the model, potentially impacting training and inference efficiency.

3. Due to the lack of stronger RAG baselines, the paper mainly focuses on comparing DecoupledRAG with the vanilla RAG setup (Table1m Figure5), which clearly lacks comprehensiveness.

4. The paper does not extensively validate whether the proposed DecoupledRAG framework can generalize across different types of LLMs and a variety of tasks.

**Questions:**

1. How does DecoupledRAG perform across different types of LLMs and a variety of NLP tasks?

2. What are the computational resource requirements for deploying DecoupledRAG, particularly when dealing with large datasets? Providing TFLOPs of DecoupledRAG is necessary.

3. Please provide more results of strong RAG baselines.

**Reviewer Confidence:**

4: The reviewer is certain that the evaluation is correct and very familiar with the relevant literature

**Scope:**

4: The work is relevant to the Web and to the track, and is of broad interest to the community

---

### Official Review · Reviewer_AWVe · 2024-12-01

**Novelty:** 6
**Technical Quality:** 6

**Review:**

### Summary

This paper introduces DecoupledRAG,  which decouples the external knowledge/documents from the context/prompt for LLM-based RAG frameworks. Specifically, DecoupledRAG first encodes external knowledge as cached key-value representations. For a specific session, the cached representations of the retrieved documents are aggregated as parameters to augment the generation, independent of the input context. This paradigm, accompanied by the low-ranking training techniques, resolves the problem of training efficiency and the long-context modeling problem for RAG, which is supported by abundant experiments.

### Strengths

1. Significance and novelty. The proposed framework can facilitate the application of RAG techniques as it solves the training efficiency and long-context problems of the Vanilla RAG pipeline. The method is novel and would make an impact on the community.
2. Clarity. The paper is well-structured and easy to follow.
3. Plenty of experiments are provided. Necessary evaluations are available to support the effectiveness. The result of the long context and efficiency test is impressive.

### Weakness

1. Storage efficiency is not discussed.  DecoupledRAG heavily relies on the offline caching of encoded external knowledge, i.e., key-value representations for all layers of all documents. This requires plenty of extra storage space compared with the Vanilla RAG. I am curious about the solution to alleviating the storage complexity and the experimental comparison with Vanilla RAG at this point.
2. Experimental details. Some experimental details are missing, e.g., the statistics of used datasets. How will the storage complexity change when the number of documents within the dataset is larger?
3. Ablation is not provided. No ablation studies are available. I wonder which components are necessary for DecoupledRAG and whether some are replaceable.

**Questions:**

Please refer to the weakness in the review.

**Reviewer Confidence:**

3: The reviewer is confident but not certain that the evaluation is correct

**Scope:**

4: The work is relevant to the Web and to the track, and is of broad interest to the community

---

### Official Review · Reviewer_j7Mo · 2024-12-01

**Novelty:** 4
**Technical Quality:** 3

**Review:**

This paper introduces DecoupledRAG, a novel framework that addresses the limitations of traditional Retrieval-Augmented Generation (RAG) systems by decoupling external knowledge from the context and injecting it via cross-attention.   The approach aims to mitigate issues related to long contexts, such as increased inference latency and degradation of fundamental capabilities, while also being more robust to the permutation of knowledge.   The framework is shown to maintain high efficiency and achieve superior performance across multiple datasets.

Weaknesses
- The innovation is somewhat limited, and the authors should reiterate their motivations.
- In longer contexts (e.g. 32k, 128k, or more), whether this method still hold up.
- The model structure is rather complex.   It's worth inquiring if the inference speed has been tested and compared with baselines.   Additionally, the time complexity could be computed.

**Questions:**

Weaknesses

**Reviewer Confidence:**

3: The reviewer is confident but not certain that the evaluation is correct

**Scope:**

2: The connection to the Web is incidental, e.g., use of Web data or API